# Correlation of an Index-Lesion-Based SPECT Dosimetry Method with Mean Tumor Dose and Clinical Outcome after ^177^Lu-PSMA-617 Radioligand Therapy

**DOI:** 10.3390/diagnostics11030428

**Published:** 2021-03-03

**Authors:** Friederike Völter, Lena Mittlmeier, Astrid Gosewisch, Julia Brosch-Lenz, Franz Josef Gildehaus, Mathias Johannes Zacherl, Leonie Beyer, Christian G. Stief, Adrien Holzgreve, Johannes Rübenthaler, Clemens C. Cyran, Guido Böning, Peter Bartenstein, Andrei Todica, Harun Ilhan

**Affiliations:** 1Department of Nuclear Medicine, University Hospital, Ludwig-Maximilians-University Munich, 80331 Munich, Germany; friederike.voelter@med.uni-muenchen.de (F.V.); lena.mittlmeier@med.uni-muenchen.de (L.M.); astrid.gosewisch@med.uni-muenchen.de (A.G.); julia.brosch-lenz@med.uni-muenchen.de (J.B.-L.); franz.gildehaus@med.uni-muenchen.de (F.J.G.); mathias.zacherl@med.uni-muenchen.de (M.J.Z.); leonie.beyer@med.uni-muenchen.de (L.B.); adrien.holzgreve@med.uni-muenchen.de (A.H.); guido.boening@med.uni-muenchen.de (G.B.); peter.bartenstein@med.uni-muenchen.de (P.B.); andrei.todica@med.uni-muenchen.de (A.T.); 2Department of Urology, University Hospital, Ludwig-Maximilians-University Munich, 80331 Munich, Germany; christian.stief@med.uni-muenchen.de; 3Department of Radiology, University Hospital, Ludwig-Maximilians-University Munich, 80331 Munich, Germany; johannes.ruebenthaler@med.uni-muenchen.de (J.R.); clemens.cyran@med.uni-muenchen.de (C.C.C.)

**Keywords:** radioligand therapy, ^177^Lu-PSMA, dosimetry, SPECT, mCRPC

## Abstract

Background: Dosimetry can tailor prostate-specific membrane-antigen-targeted radioligand therapy (PSMA-RLT) for metastatic castration-resistant prostate cancer (mCRPC). However, whole-body tumor dosimetry is challenging in patients with a high tumor burden. We evaluate a simplified index-lesion-based single-photon emission computed tomography (SPECT) dosimetry method in correlation with clinical outcome. Methods: 30 mCRPC patients were included (median 71 years). The dosimetry was performed for the first cycle using quantitative ^177^Lu-SPECT. The response was evaluated using RECIST 1.1 and PERCIST criteria, as well as changes in PSMA-positive tumor volume (PSMA-TV) in post-therapy PSMA-PET and biochemical response according to PSA changes after two RLT cycles. Results: Mean tumor doses as well as index-lesion doses were significantly higher in PERCIST responders compared to non-responders (10.2 ± 12.0 Gy/GBq vs. 4.0 ± 2.9 Gy/GBq, *p* = 0.03 and 13.7 ± 14.2 Gy/GBq vs. 5.9 ± 4.4 Gy/GBq, *p* = 0.04, respectively). No significant differences in mean tumor and index lesion doses were observed between responders and non-responders according to RECIST 1.1, PSMA-TV, and biochemical response criteria. Conclusion: Compared to mean tumor doses on a patient level, single index-lesion-based SPECT dosimetry correlates equally well with the response to PSMA-RLT according to PERCIST criteria and may represent a fast and feasible dosimetry approach for clinical routine.

## 1. Introduction

Prostate-specific-membrane-antigen (PSMA) targeted theranostic approaches for metastatic castration-resistant prostate cancer (mCRPC) are of rising clinical importance [1]. PSMA-targeted radioligand therapy (PSMA-RLT) using β- and α-emitters, such as ^177^Lu- and ^225^Ac-labeled PSMA-617 and PSMA-I&T, allows selective irradiation of PSMA-expressing prostate cancer (PC) cells and is used in many centers worldwide [2,3,4,5,6,7,8] and results of the first prospective phase 3 trial (VISION; NCT03511664) are anticipated [9].

Currently, PSMA-RLT represents a last-line therapy option in advanced mCPRC after exhaustion of approved therapy options, including anti-hormonal therapy, novel anti-hormonal agents, such as abiraterone acetate and/or enzalutamide, and taxane-based chemotherapy [10]. According to a recent meta-analysis on the efficacy and safety of ^177^Lu-PSMA RLT, including 17 articles and 744 patients, biochemical response with a prostate-specific antigen (PSA) decline >50% is observed in 46% (95% CI: 40–53%) despite progressive disease after exhaustion of approved therapy options [11]. Nonetheless, not all patients will respond to PSMA-RLT, and biochemical progression is observed in up to 37% of patients (95% CI: 34–40%) despite sufficient PSMA-expression in pre-therapeutic positron emission tomography (PET) imaging [11]. Low intra-tumoral doses during ^177^Lu-PSMA RLT might represent one of the reasons for therapy failure despite high PSMA-expression of lesions in PSMA PET imaging. PSMA-RLT is a trade-off between sufficient tumor dose while maintaining recommended dose limits for organs at risk, including the kidneys, bone marrow, and salivary glands, which underlines the clinical importance of dosimetry during RLT [12]. However, full individual dosimetry with the segmentation of every single tumor lesion is challenging, especially in patients with a high tumor burden. Recently, Violet et al. presented data on the correlation of whole-body tumor-absorbed doses after ^177^Lu-PSMA-617 RLT with PSA response [13]. The median whole-body tumor dose was 14.1 Gy in patients with PSA decline ≥ of 50% compared to patients with a PSA decline of less than 50% with a median dose of only 9.6 Gy, providing a rationale for individualized dosimetry during PSMA-RLT. However, tumor dosimetry, including all tumor lesions or multiple selected lesions, is elaborate and time-consuming during a clinical routine, particularly in the case of single-photon emission computed tomography (SPECT)-based dosimetry with complex tumor delineation when performed manually. This study aimed to evaluate and compare an easy and feasible, index-lesion-based SPECT dosimetry method with mean tumor doses in multiple tumor lesions on a patient level and to correlate dosimetry data with clinical outcome after ^177^Lu-PSMA RLT.

## 2. Materials and Methods

### 2.1. Patients

For this retrospective analysis, we included patients with mCRPC who received 2 cycles of ^177^Lu-PSMA-617 radioligand therapy at our institution between September 2014 and June 2018. All patients received PSMA-targeted PET combined with diagnostic computed tomography (CT) at our institution to assess PSMA-expression prior to RLT and for follow-up after two cycles of PSMA-617 RLT. Patients with different ligands for PSMA-RLT (e.g., PSMA-i&T) were excluded to improve the comparability of dosimetry data. Patients without quantitative SPECT imaging at least three time points (24, 48, 72 h postinjection), patients receiving only one therapy cycle were excluded from the analysis. Figure 1 provides a flowchart summarizing the criteria for patient inclusion. Indication for PSMA RLT was in accordance with the procedure guidelines for radionuclide therapy with ^177^Lu-labeled PSMA-ligands of the European Association of Nuclear Medicine (EANM) and with recommendations of the German Society of Nuclear Medicine [10,14]. Furthermore, indication for radioligand therapy was confirmed by the local interdisciplinary tumour board. RLT was performed in accordance with the German Medical Products Act (AMG) §13.2b and the updated declaration of Helsinki, § 37 (Unproven Interventions in Clinical Practice). All patients were informed about the experimental nature of this unapproved therapy as well as possible risks and side effects and provided written informed consent. The study was approved by the local ethics committee (approval number 20-178, approval date: 02.02.2019).

### 2.2. ^177^Lu-PSMA-617 Therapy

^177^Lu-PSMA-617 was obtained by radiolabeling of DOTA-PSMA-617 precursor (ABX GmbH, Radeberg, Germany) with no-carrier-added ^177^LuCl_3_ (ITG GmbH, Garching, Germany) as described previously [15]. ^177^Lu-PSMA-617 was injected intravenously over 15–20 min using an infusion pump. Before and after therapy, all patients were hydrated with 2 liters of sodium-iodine. During and after therapy, all patients received cool packs to reduce the blood flow of the salivary glands.

### 2.3. PET/CT Imaging

Radiolabeling and administration of ^68^Ga-PSMA-11 and 18F-PSMA-1007 were performed according to previously reported radiosynthesis and administration procedures [16,17]. Immediately after injection of the tracer, 20 mg of furosemide was administered intravenously. Patients were asked to empty the bladder prior to the scan. PET/CT scans were acquired 60 min after tracer injection using an iodine-containing contrast agent (Ultravist 300, Bayer Pharma AG, Berlin, Germany; or Imeron 300, Bracco, Konstanz; Germany 2.5 mL/s) during the portal venous phase. For reconstruction, the TrueX algorithm (3 iterations, 21 subsets; Biograph 64, Siemens Healthineers, Erlangen, Germany) or the VUE Point FX algorithm (2 iterations, 36 subsets; Discovery 690) was used with an axial 168 x 168 matrix. Phantom studies based on the National Electrical Manufacturers Association NU2-2001 standard were conducted to allow valid pooling of results between different scanners. Mean injected activity was 213.1 ± 49.7 MBq in pre- and 212.6 ± 45.1 MBq in post-therapeutic PET.

### 2.4. Dosimetry

Dosimetry was performed using quantitative ^177^Lu-SPECT scans of the abdomen 24, 48, and 72 h postinjection. The images were acquired over 15 min on a dual-headed Symbia T2 SPECT/CT system (Siemens Healthineers, Erlangen, Germany) as described previously [12,15]. For attenuation correction, a low-dose CT scan (AC-CT) was acquired together with the first SPECT scan 24 h postinjection. The AC-CT was coregistered on the SPECT scans acquired 48 and 72 h after therapy (rigid body co-registration, PMOD Version 3.609, PMOD Technologies, Zurich, Switzerland). Quantitative SPECT reconstruction employed an in-house maximum a posteriori (MAP) algorithm with 20 iterations, 16 subsets, and a penalty factor of 0.001, as described previously [8,10]. Scatter correction and resolution decompensation was applied in addition to the mentioned attenuation correction. For each patient, three to five lesions with the highest visual uptake in abdominal SPECT were evaluated. Tumor lesions were segmented semi-manually by placing volumes of interest (VOIs) with an iso-contour of 40% as proposed by Collarino et al. and confirmed by in-house phantom studies [18]. Lesion dosimetry was based on a mono-exponential fit model and mass-scaled sphere *S*-values (Figure 2).

### 2.5. Response Assessment

Biochemical response assessment was based on PSA-levels according to previously described protocols [19,20,21]. The complete biochemical response was defined as a non-measurable PSA level (0 ng/mL) after PSMA RLT, partial biochemical response (PR) defined a PSA-decline ≥50%, stable disease (SD) as a PSA change between −50% and +25%, and progressive disease (PD) as a PSA increase ≥25%.

Radiographic response assessment was performed according to RECIST 1.1 and modified PERCIST criteria [22,23]. RECIST 1.1 evaluation was performed on diagnostic CT images acquired during follow-up PET/CT imaging with target lesions defined as ≥ 1 cm for soft tissue lesions and ≥1.5 cm in shortest axis for lymph node metastases. The sum of diameter (SOD) of up to five target lesions with a maximum of two lesions per organ was evaluated. Sclerotic bone metastases were considered non-measurable lesions. PR was defined as a decrease of ≥30% in SOD, and PD as an increase of ≥20%. A SOD change between a decrease of less than 30% and an increase of less than 20% was considered SD. In the case of bone-only disease, SD was considered for equivocal disease or no change. The appearance of new lesions was considered PD. For PERCIST evaluation, background activity was determined using the standardised uptake value (SUV) of a 3 cm diameter spherical volume of interest placed in the right side of the liver. SUVpeak of the single hottest tumor lesion was evaluated manually using a 1 mL VOI. The hottest tumor lesion in pre- and post-therapeutic PET did not need to be the same according to PERCIST criteria defined for FDG-PET. PR was defined as a decrease of SUVpeak of the hottest tumor lesion of 30% and a minimum of 0.8 SUV units [22]. PD was defined as an increase of SUVpeak of 30% or the occurrence of new tumor lesions. SD was defined as a decrease or increase of SUVpeak of the hottest tumor lesion between −30 and +30% without the occurrence of new metastases.

Furthermore, changes in PSMA-positive tumor volume (PSMA-TV) after PSMA-RLT were evaluated [21,24]. Pre- and post-therapeutic ^68^Ga-PSMA-11 PET/CT was available in 21 patients; the remaining 9 patients with 18F-PSMA-1007 PET/CT were excluded from this analysis due to different tracer biodistribution [17]. A lower threshold was defined as the mean SUV within a circular reference VOI of 2 cm diameter placed in the liver, avoiding the inclusion of major intrahepatic vessels based on contrast-enhanced CT images. Whole tumor volume was evaluated using a threshold-based segmentation algorithm included in Hermes Affinity v1.1 (Hermes Medical Solutions, Stockholm, Sweden), followed by manual correction for unspecific tracer uptake or organs with non-tumor related tracer uptakes, such as the kidneys, spleen, salivary glands, gut, and bladder. Cutoff values for therapy response in PSMA-TV assessment were defined according to previously published data [21,24]. A decline of ≥30% was defined as a partial response; stable disease was defined as a change of whole tumor volume between −30% and +30%. PD in PSMA-TV assessment was defined as an increase of ≥30%.

### 2.6. Statistical Analysis

Statistical analysis was performed using SPSS Statistics 24 (IBM, New York, NY, USA) and GraphPad Prism, Version 8.4.3 (GraphPad Software, San Diego, CA, USA). Patients were divided into subgroups based on treatment response. Estimated lesion absorbed doses were compared to therapy outcome, which was determined based on image response assessment using PERCIST criteria, RECIST 1.1 criteria, and PSMA-TV applied on pre- and post-therapeutic ^68^Ga-PSMA PET/CT as well as biochemical PSA-response. Patients were grouped as non-responders in the case of SD and PD and as responders in the case of CR or PR. Gaussian distribution was evaluated using the Shapiro–Wilk test. For analysis of variance between subgroups, Mann–Whitney *U* test was used. Receiver operating characteristic analysis (ROC analysis) was performed for mean dose and the dose of index-lesions in therapy response evaluations. The cutoff doses showing the maximum sum of sensitivity and specificity in ROC analysis were selected as threshold doses. The frequency of responders and non-responders above and below the determined threshold value were compared using the chi-squared test.

## 3. Results

### 3.1. Patient Characteristics

Thirty patients with mCRPC (mean age of 71.4 ± 9.9) were included. Detailed patient characteristics are provided in Table 1. Median injected activity for the first and second therapy cycle were 6.01 GBq ± 0.8 and 6.00 GBq ± 0.9, respectively.

### 3.2. Biochemical Response Assessment

Median PSA decline after two PSMA-RLT cycles was −37.4% (range: −99.6 to 730.3% compared to baseline PSA). 13/30 (43%) patients were categorized as responders and showed a PSA decline ≥ of 50% (PR), the remaining 17 patients (57%) were grouped as non-responders, with 9 patients (30%) showing a PSA change between ≤−50% and +25% (SD) and 8 patients (27%) showing a PSA increase ≥25% (PD).

### 3.3. Therapy Response Assessment Using PERCIST and RECIST 1.1 Criteria

According to RECIST 1.1, PR was observed in 2/30 (7%), SD in 8/30 (26%), and PD in 20/30 patients (67%). According to PERCIST, PR was observed in 10/30 (33%) and PD in 20/30 patients (67%). No patient showed SD according to PERCIST criteria. Response assessment evaluating PSMA-TV showed PR in 7/21 (33%), SD in 6/21 (29%), and PD in 8/21 patients (38%).

Comparing the results of PERCIST and RECIST evaluations, for 21 of 30 patients (70%), PERCIST and RECIST matched. In 7/9 patients with a mismatch between PERCIST and RECIST, a significant reduction of PSMA expression in skeletal target lesions resulted in PR according to PERCIST criteria, while bone metastases were non-measurable according to RECIST without unequivocal progression resulting in SD. One patient showed an increasing SUVpeak of bone metastases resulting in PD according to PERCIST, while the RECIST evaluation showed SD. Another patient with mismatching results in PERCIST and RECIST evaluation showed a significantly growing cardiophrenic lymph node metastasis, while SUVpeak of this metastasis was stable and other lymph node metastases were also significantly decreasing in size and PSMA-expression, resulting in PR according to PERCIST and PD according to RECIST 1.1 (Figure 3).

Comparing PSMA-TV with PERCIST and RECIST criteria, response assessment using PSMA-TV showed matching results with PERCIST and RECIST 1.1 evaluation in 14/21 (67%) and 11/21 (52%) patients, respectively. A detailed comparison of radiographic response according to different criteria is provided in Table 2.

### 3.4. Comparison of Image-Based and -Biochemical Response Evaluation

Results of radiographic response assessment and biochemical response evaluation are listed in detail in Table 3. According to PERCIST criteria, there was a good correlation with biochemical response assessment in 17/30 patients (57%), with 8 patients showing PD and 9 PR. PD, according to PERCIST, despite PR in biochemical response assessment, was observed in three patients due to the occurrence of new metastases and rising SUVpeak in one patient. Mean PSA decline in patients with PR according to PERCIST criteria was 81.0 ± 21.3% compared to a mean increase of 85.2 ± 229.5% in patients with PD (*p* < 0.001). RECIST evaluation was in concordance with biochemical response assessment in 11/30 (37%) cases, including 8 patients with PD, one with SD, and two with PR. PSMA-TV response assessment revealed concordant biochemical response in 12/21 patients (57%). While patients with PR showed a PSA decline of 73.4 ± 24.6%, patients with PD showed a PSA increase of 147.7 ± 286.4% (*p* < 0.01).

### 3.5. Dosimetry Results

Dosimetry was performed for multiple tumor lesions as a correlate for mean tumor dose on a patient-level in a total of 141 tumor lesions (97 bone metastases, 44 lymph node metastases, median 5 lesions per patient). The mean tumor dose was 5.7 ± 6.4 Gy/GBq with a mean absorbed dose of 7.7 ± 9.7 Gy/GBq and 4.7 ± 3.9 Gy/GBq in lymph node and bone metastases, respectively. The mean index-lesion-based dosimetry was calculated for single tumor lesions on a patient level. The mean dose in the index-lesion-based method was 8.5 ± 9.4 Gy/GBq.

#### 3.5.1. Dose Estimations of Biochemical Responders and Non-Responders

There was no significant difference regarding the lesion absorbed doses in biochemical responders (13/30 patients, 43%) and non-responders (17/30 patients, 57%) with a mean tumor dose of 8.3 ± 10.1 Gy/GBq vs. 4.3 ± 3.3 Gy/GBq (*p* = 0.21; Figure 4a). Similarly, no significant differences were observed for index-lesion dosimetry (biochemical responders: 11.7 ± 12.9 Gy/GBq vs. biochemical non-responders: 6.1 ± 4.7 Gy/GBq; *p* = 0.12; Figure 4b).

#### 3.5.2. Dose Estimations for Responders and Non-responders as Evaluated via PERCIST, RECIST and PSMA-TV

The averaged mean tumor dose was significantly higher in responders according to PERCIST criteria (10/30; patients 33%) compared to non-responders (20/30 patients; 67%) with 10.2 ± 12.0 Gy/GBq vs. 4.0 ± 2.9 Gy/GBq, *p* = 0.03 (Figure 5a). According to RECIST 1.1, no significant differences were observed between responders and non-responders (25.0 ± 25.5 Gy/GBq vs. 4.7 ± 3.3 Gy/GBq, *p* = 0.11); however, only 2/30 patients were grouped as responders, whereas 28/30 patients showed SD and PD (Figure 5b). Responders according to PSMA-TV evaluation (7/21 patients 33%) had a mean averaged tumor dose of 10.7 ± 14.5 Gy/GBq compared to non-responders (14/21 patients 67%) with a mean tumor dose of 4.5 ± 3.6 Gy/GBq (*p* = 0.32) (Figure 5c).

Similar results were observed for the index-lesion-based analysis with significantly higher tumor doses in hottest lesions of responders according to PERCIST criteria (mean: 13.7 ± 14.2 Gy/GBq) compared to non-responders (mean: 5.9 ± 4.4 Gy/GBq; *p* = 0.04; Figure 6a). Responders and non-responders according to RECIST 1.1 criteria and PSMA-TV assessment did not show significant differences with mean index-lesion absorbed doses of 30.1 ± 30.6 Gy/GBq vs. 7.0 ± 4.9 Gy/GBq (*p* = 0.11) and 13.8 ± 17.0 Gy/GBq vs. 6.6 ± 5.3 Gy/GBq (*p* = 0.29), respectively (Figure 6b,c).

ROC-analysis showed a significant C-index for mean tumor volume dose and index-lesion dose in responders and non-responders according to PERCIST criteria (0.75 and 0.86; *p* = 0.03 and *p* = 0.04, respectively; Figure 7). Threshold values of mean dose and the dose of the index-lesions were 5.0 Gy/GBq and 5.8 Gy/GBq according to PERCIST and 6.5 and 8.0 Gy/GBq according to RECIST. The frequency of therapy response was significantly higher in patients with mean and index-lesion tumor doses above the determined threshold values compared to patients with lower doses (PERCIST: 16.7% vs. 58.3% for mean tumor volume dose and 8.3% vs. 50% for index tumor lesion dose) and RECIST analysis (0% vs. 20% for mean tumor volume dose and index tumor lesion dose, respectively).

## 4. Discussion

Novel therapeutic agents from chemotherapy to second-line anti-hormonal agents and radioligand therapy using ^177^Lu- or ^225^Ac-labeled PSMA ligands changed the landscape of available therapy options for mCRPC [25]. Results of the first phase 3 trial comparing ^177^Lu-PSMA-617 with best supportive care in mCRPC patients after taxane-based chemotherapy (VISION; NCT03511664) might expand the armamentarium for mCRPC with a novel approved radiopharmaceutical besides ^223^Radium-dichloride (^223^Ra) [9,26]. However, unlike ^223^Ra, which targets bone metastases only, ^177^Lu-PSMA-ligands represent a systemic therapy option targeting all PSMA-avid tumor lesions. Theranostic concepts for prostate cancer include pre-therapeutic visualization and verification of PSMA-expression prior to RLT by PSMA-PET, as well as pre- and peri-therapeutic dosimetry [1]. However, the clinical implementation of individualized dosimetry for prostate cancer theranostics is still a matter of debate, and it remains unclear whether dosimetry data will improve patient management, especially when considering additional requirements and efforts, including standardization and harmonization of therapy and dosimetry protocols [27,28]. Furthermore, it is mandatory to evaluate dosimetry results not only in relation to RLT associated toxicity but also with therapy outcome.

In this study, we evaluate dosimetry for the first RLT cycle using ^177^Lu-PSMA-617 in correlation with outcome parameters, including radiographic response in PSMA-PET and diagnostic CT according to RECIST 1.1, PERCIST, and changes in PSMA-TV as well as biochemical response according to changes in PSA values. We observed PSA decline ≥50% in 43% of patients after 2 PSMA-RLT cycles with a median cumulative activity of 12 GBq. These results are comparable to other groups as demonstrated in a meta-analysis summarizing 10 studies published between 2015 and 2018, including a total of 455 patients with 34.5% showing a PSA decline ≥of 50% after PSMA-RLT. Several groups demonstrated the feasibility of response assessment based on PSMA-PET-based parameters in correlation with PSA-response, including modified PERCIST as well as PSMA-TV after RLT and/or chemotherapy [21,24], whereas response assessment in mCRPC using RECIST 1.1 criteria is hampered by non-measurable disease in case of frequently observed bone metastases [29,30]. We observed similar results in our cohort with higher concordance rates between biochemical response and PSMA-PET-based criteria as opposed to morphological RECIST 1.1 criteria (Table 3). Despite PERCIST being validated for FDG-PET only, a concordance rate of 57% indicates that PERCIST criteria might also be applicable for response assessment in prostate cancer. Nonetheless, larger cohorts and preferably prospective trials are necessary to confirm this hypothesis. Similar to PERCIST, a concordance rate of 57% was also observed between biochemical response and response according to PSMA-TV, whereas the correlation rate between biochemical response and response according to RECIST 1.1 criteria was only 37%. The relevance of molecular PET parameters in prostate cancer was also highlighted in a recently published consensus statement on PSMA PET/CT-based response assessment, emphasizing the value of semi-quantitative PET parameters, such as standardized uptake value (SUV) and quantification of PSMA-avid tumor burden [31]. Nonetheless, the discordance of biochemical response and radiographic response according to PERCIST and PSMA-TV was observed in 43% of our cohort, respectively. PERCIST showed PD in 12 patients despite biochemical SD in 8 and PR 4 patients. One of the main reasons for this discrepancy is the fact that all new lesions are considered as PD according to PERCIST criteria [22]. However, this approach might not be appropriate in mCRPC and RLT, where the response is observed in PSMA-avid lesions, and new lesions, particularly with small volume, might not impact patient outcome [32]. These patients can be considered as mixed responders [32]. However, the mixed response was not included in the present analysis considering our relatively small cohort of 30 patients. Nonetheless, these considerations must be taken into account when evaluating dosimetry data in correlation with response to RLT. The ideal metric for response assessment after RLT is yet to be defined, particularly as serum PSA can be limited in patients receiving androgen-deprivation therapy. Furthermore, for radionuclide therapy, using ^223^Ra PSA is of limited prognostic value in contrast to changes in alkaline phosphatase and lactate dehydrogenase levels [33]. Thus, radiographic response assessment implementing PSMA-based semi-quantitative parameters will gain increasing relevance in mCRPC. This includes therapy-associated changes in PSMA-avid tumor volume, which has been demonstrated for ^68^Ga-labeled PSMA ligands [21]. Nine patients with 18F-PSMA-1007 PET have been excluded from PSMA-TV analysis, as different biodistribution will affect tumor delineation using cutoff values established for ^68^Ga-PSMA ligands. Further trials investigating different PSMA-ligands labeled with ^68^Ga and 18F, even, including phantom measurements for pooling of different tracers and scanners, are needed to further establish the value of PSMA-PET during follow-up after RLT.

Recently, Violet et al. presented encouraging data on the value of SPECT dosimetry in 30 patients receiving up to four cycles of ^177^Lu-PSMA-617 within a prospective trial [13]. SPECT-based tumor dosimetry was performed for the whole tumor burden and correlated with PSA response at 12 weeks with a significantly higher mean dose for “whole-body” tumor lesions in patients with PSA decline ≥50% compared to patients with a decline <50% (14.1 Gy vs. 9.6 Gy; *p* < 0.01). This was not observed when considering index-lesion tumor dose (numbers not shown; *p* = 0.09). In the present analysis, dosimetry was performed with two different approaches on a patient level to evaluate a clinically feasible and applicable, time-saving method for peri-therapeutic RLT dosimetry during daily routine. The dosimetry was performed for multiple lesions per patient to assess the mean tumor dose as well as for a single index-lesion with the highest uptake on ^177^Lu-PSMA-617 SPECT. In contrast to Violet et al., we did not observe significant differences between dosimetry data of biochemical responders and non-responders (Figure 4) with a mean tumor dose of 8.3 Gy/GBq for responders and 4.3 Gy/GBq for non-responders (*p* = 0.21). Similar results were observed for index-lesion-based dosimetry (11.7 Gy/GBq vs. 6.1 Gy/GBq; *p* = 0.12). These contradictory results might be explained by differences in the patient population and therapy protocols. Violet et al. excluded patients with single FDG-avid tumor lesions with concomitant low PSMA-expression. FDG-PET in mCRPC patients is not routinely performed prior to RLT at our site. Therefore, we cannot rule out whether patients with FDG-avid tumor lesions were included or not. However, in agreement with other groups, we believe that PSMA-RLT might still be successful in patients with PSMA-avid disease despite single FDG-avid lesions [34]. Furthermore, Violet et al. performed up to four RLT cycles with a mean activity of 7.8 GBq, whereas we performed two cycles with a mean activity of 6 GBq per cycle. It remains unclear whether higher therapy doses and cumulative activities would have resulted in higher differences regarding dosimetry data of biochemical responders and non-responders in our cohort. Nonetheless, Rathke et al. demonstrated a tendency for a positive dose–response relationship in mCRPC patients treated with activities between 4, 6, 7.4, and 9.3 GBq ^177^Lu-PSMA-617 with higher rates of partial remission in patients receiving higher treatment activities, whereas initial PSA response showed no correlation with treatment activity [35]. This also indicates that initial PSA response at a predefined time point might not be the ideal metric for response assessment after RLT, which is also confirmed by the fact that delayed PSA-response can be observed in up to 29% when performing a second or third RLT cycle [3]. Nonetheless, it remains important to identify non-responders at early-stages in order not to delay alternative therapy options. When considering response according to PERCIST criteria, we could demonstrate that dosimetry for the first therapy cycle correlates with response to RLT (Figure 5 and Figure 6). Both mean tumor dose, as well as index-lesion-based tumor dose, was significantly higher in responders compared to non-responders. Furthermore, ROC-analysis provided cutoff values for successful therapy. Despite having analyzed differences in tumor dose between responders and non-responders according to RECIST 1.1 criteria, the interpretation of these results is highly limited due to the very low number of responders (2/30 patients, 7%), which once again emphasizes that RECIST 1.1 does not represent the best response criteria after PSMA-RLT.

Nonetheless, considering our results for PERCIST responders and non-responders, index-lesion-based dosimetry might represent a feasible and simple dosimetry approach for clinical routine and has the potential to impact patient management as early as at the time of the first therapy cycle. However, PSA-changes and clinical factors, including quality of life, still have to be taken into account when considering changes in management. Furthermore, dosimetry data for single-lesions has to be interpreted with caution, especially when considering the impact of heterogeneity and variance of single lesions on dosimetry with an uncertainty ranging from 14% to 102% for single index-lesions [36], which also represents one of the main limitations of our analysis. Further limitations include the retrospective design of this study and the relatively small sample size despite having performed PSMA-RLT in a considerably larger cohort. However, patients without SPECT data at least three time points after RLT, as well as patients where RLT was performed using a different PSMA ligand (e.g., PSMA-I&T), were excluded for improved comparison and interpretation of our data.

## 5. Conclusions

Peri-therapeutic dosimetry during the first cycle of PSMA-RLT correlates with therapy response according to PERCIST criteria. Both mean tumor dose for multiple lesions on a patient level and single index-lesion-based SPECT dosimetry show significantly higher dose values in responders compared to non-responders indicating that the index-lesion-based dosimetry method may represent a fast and feasible dosimetry method during the clinical routine.

## Figures and Tables

**Figure 1 diagnostics-11-00428-f001:**
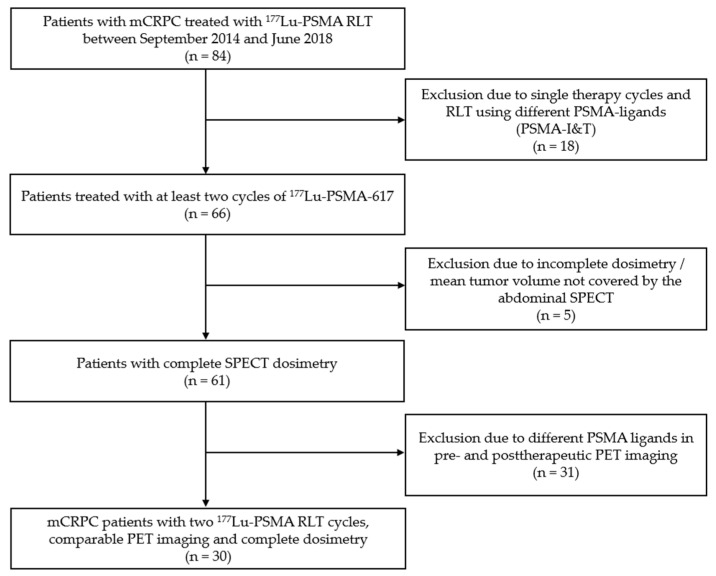
Flowchart with detailed information on patient selection.

**Figure 2 diagnostics-11-00428-f002:**
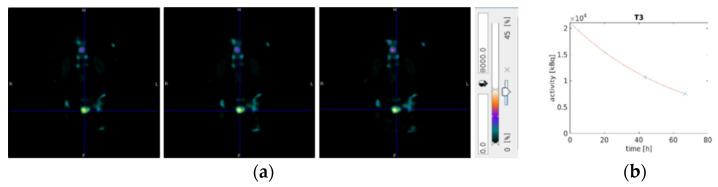
(**a**) Single-photon emission computed tomography (SPECT)/diagnostic computed tomography (CT) images at 24, 48 and 72 h after therapy with semi-manually placed the volume of interest (VOI) in a single bone lesion (green circle); (**b**) Mono-exponential fit curve of radioactivity over time of the lesion delineated in (**a**).

**Figure 3 diagnostics-11-00428-f003:**
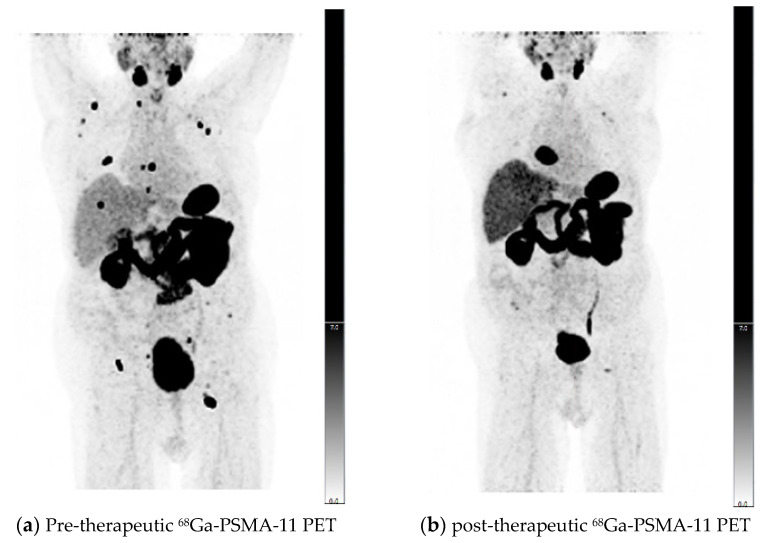
Pre- (**a**) and post-therapeutic PSMA-11 PET/CT (**b**) of an mCRPC patient with metastatic disease in bones and lymph nodes. Follow-up was performed after two cycles of ^177^Lu-PSMA-617 RLT. Partial response was observed according to PERCIST criteria, and progressive disease according to RECIST 1.1 due to the significantly increased size of a cardiophrenic lymph node metastasis, which was selected as a measurable target lesion in pre-therapeutic CT. SPECT dosimetry showed a mean dose of 5.4 Gy/GBq and a dose of 6.2 Gy/GBq for the index-lesion with the highest uptake.

**Figure 4 diagnostics-11-00428-f004:**
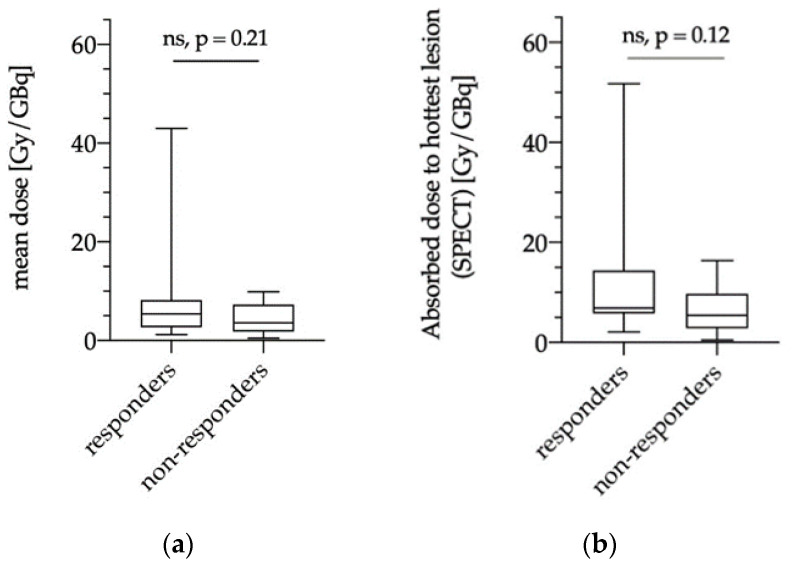
Comparison of mean absorbed doses in biochemical responders and non-responders using SPECT dosimetry for averaged tumor lesions on a patient-level (“mean dose”, (**a**)) and for a single lesion with highest uptake (“hottest lesion”, (**b**)) in the index-lesion-based dosimetry method. ns = not significant.

**Figure 5 diagnostics-11-00428-f005:**
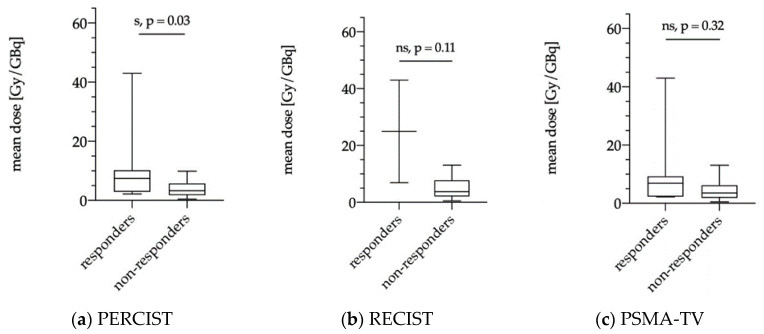
Mean absorbed tumor dose for averaged multiple lesions (“mean dose” in Gy/GBq) stratified according to responders and non-responders using modified PERCIST criteria (**a**), RECIST 1.1 criteria (**b**), and PSMA-TV (*n* = 21) (**c**) evaluated. s = significant, ns = not significant.

**Figure 6 diagnostics-11-00428-f006:**
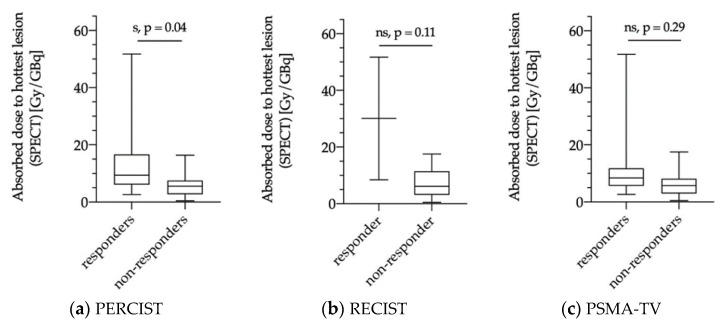
Index-lesion-based dosimetry for single tumor lesions with the highest uptake (“hottest lesion” in Gy/GBq) stratified according to responders and non-responders using modified PERCIST criteria (**a**), RECIST 1.1 criteria (**b**), and PSMA-TV (*n* = 21) (**c**). s = significant, ns = not significant.

**Figure 7 diagnostics-11-00428-f007:**
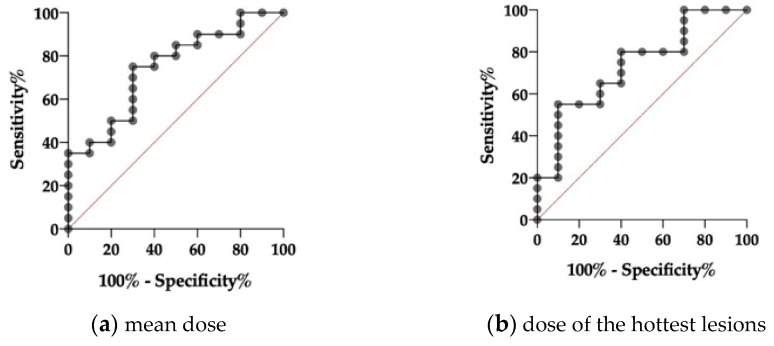
Receiver operating characteristics curve of averaged tumor lesions (“mean dose”, (**a**)) and the index-lesion-based analysis (“hottest lesion”, (**b**)) for responders and non-responders according to PERCIST criteria mean with a C-index of 0.75 and a *p*-value of 0.03 for the mean dose and a-with-a C-index of 0.86 and a *p*-value of 0.04 for the hottest-lesion dosimetry, respectively.

**Table 1 diagnostics-11-00428-t001:** Initial clinical characteristics of the included patients.

Initial Clinical Characteristics	Total Patients = 30	
**Age (years)**		
Median (range)	71.4	(52.4–91.2)
**Previous therapies (*n*)**		
Radical prostatectomy	23	(77%)
Radiotherapy	19	(63%)
Androgen deprivation therapy	27	(90%)
Brachytherapy	1	(3%)
Abiraterone/enzalutamide	19	(63%)
Radium-223	9	(30%)
Chemotherapy (docetaxel and/or cabazitaxel)	15	(50%)
**Serum PSA baseline (ng/mL)**		
Median (range)	129.0	(1.4–9237.0)
**Gleason score (*n* = 23)**		
Median (range)	9	(6–10)
**Metastases localization (*n*)**		
Bone	27	(90%)
Lymph nodes	23	(77%)
Liver	3	(10%)
Lungs	2	(7%)
Local recurrence	5	(17%)
**Interval baseline PET–1st therapy cycle**		
Median (range) (weeks)	4.4	(0.6–16.7)
**Interval 1st–2nd therapy cycle**		
Median (range) (weeks)	8	(4.3–10.1)
**Interval 1st therapy cycle–follow-up PET**		
Median (range) (weeks)	17.1	(13.1–22.2)

**Table 2 diagnostics-11-00428-t002:** Radiographic response assessment evaluating prostate-specific membrane-antigen-targeted radioligand therapy (PSMA-TV) in 21 patients compared to PERCIST and RECIST 1.1 evaluation. PD: progressive disease; SD: stable disease; PR: partial response.

		PSMA-TV		
		PD	SD	PR
**PERCIST**	PD	8 (38%)	3 (14%)	1 (5%)
	SD	0 (0%)	0 (0%)	0 (0%)
	PR	0 (0%)	3 (14%)	6 (29%)
**RECIST 1.1**	PD	7 (33%)	4 (19%)	1 (5%)
	SD	1 (5%)	2 (10%)	4 (19%)
	PR	0 (0%)	0 (0%)	2 (10%)

**Table 3 diagnostics-11-00428-t003:** Results of biochemical response assessment and image response assessment evaluated using PERCIST criteria in PET and RECIST criteria in CT in 30 patients.

		PERCIST			RECIST			PSMA-TV		
		PD	SD	PR	PD	SD	PR	PD	SD	PR
	PD	8 (27%)	0 (0%)	0 (0%)	8 (27%)	0 (0%)	0 (0%)	4 (19%)	1 (5%)	0 (0%)
PSA	SD	8 (27%)	0 (0%)	1 (3%)	8 (27%)	1 (3%)	0 (0%)	3 (14%)	2 (10%)	1 (10%)
	PR	4 (13%)	0 (0%)	9 (27%)	4 (13%)	7 (27%)	2 (7%)	1 (5%)	3 (14%)	6 (29%)

note: PSMA-TV was evaluated for 21 patients.

## Data Availability

The data presented in this study can be made available from the corresponding author upon reasonable request.

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
