# Peer review of "Correlation of an Index-Lesion-Based SPECT Dosimetry Method with Mean Tumor Dose and Clinical Outcome after 177Lu-PSMA-617 Radioligand Therapy"

_diagnostics, 2021, doi:10.3390/diagnostics11030428_

Round 1

Reviewer 1 Report

In the present paper, Völter and colleagues investigated a single index-lesion based SPECT dosimetry method which correlated equally well with response to PSMA-RLT. It has been concluded that a fast and feasible dosimetry approach for clinical routine has been presented.

Major issues:

30 mCRPC patients seems to be rather low. Nonetheless, significant results were gained. In the field of NET, up to 1,000 PRRT patients have been investigated in a single-center study. Thus, in comparison, the enrolled number of subjects in the present study is rather low. Please comment on such differences between RLT and PRRT.

What was the critera to treat the patients? Did the authors used a comparable score to the established Krenning Score for NET patients treated with PRRT? It says a local tumor board has decided on RLT, but what are the intrascan criteria?

Is the isocontour of 40% established? Please comment on this.

For the SPECT assessments, three time-points have been used. Why exactly 24, 48 and 72h pi?

„PR was defined as a decrease of SUVpeak of the hottest tumor lesion of 30% and a minimum of 0.8 SUV units.“ The authors should cite a paper here why exactly 0.8 SUV has been used.

Significant differences in 68Ga vs 18 F PET for PSMA have been reported, in particular on normal biodistribution and detection rate. Consequently, 1007 scans have been removed. The authors should, however, discuss whether sophisticated approaches such as phantom studies may have lay the proper groundwork to include 18F scans as well. The authors later state that patients scanned with PSMA-I&T have also been removed (at the end of the Discussion), but this is not stated in the methods. To ensure clarification, the authors should provide a flowchart to see how many patients have been excluded due to PSMA-I&T or 1007. Have patients also been excluded due to the use of different 177Lu-labeled PSMA radiotracers? Patients may have also been excluded due to the use 225Ac-labeled PSMA. If so, this should also be part of the flowchart.

Figures 2-4: How are these massive errors bars are explained?

Is there any interferences with previous Ra-223 therapies in terms of maximum dose to organs at risk? Why have such patients treated with RLT as well? The authors may discuss that in terms of relevant hematotoxicity.

Minor issues:

Figures 1, 6: A scale bar is needed.

Please refrain from overexaggerating wording, such as “highly” in the first para of the Introduction.

Reviewer 2 Report

This is an interesting paper that strives to use a simplified dosimetry approach (based on index lesion dose) as a replacement for more complicated mean tumor dose approach. The paper is well-organized and the analytical and statistical methods of this retrospective study are very thoroughly described. The authors measure outcome after 2 courses of PSMA-RLT using a variety of approaches, and, as others have reported, found significant discrepancies among the different measures (PSA levels, RECIST, PERCIST, PSMA tumor volume, etc.). They then compute dosimetry values from [Lu-177]Lu-PSMA-617 SPECT scans and show a significant correlation between mean tumor dose and the PERCIST method of classifying outcomes, but lack of significant correlation with other outcome measures. Finally, the relation between the new dosimetry method (index lesion dose) and PERCIST response is also significant (and there is not a significant correlation to other outcome measures), suggesting the potential that these dosimetry methods can be used interchangeably. In the final paragraphs, the authors speculate about using this dosimetry information from the first course of PSMA-RLT to influence patient management, but this seems a bit of a stretch to me considering the wide variance in different outcome measures (i.e. which actually reflects patient improvement in health?), high uncertainties in index lesion doses, and also the fact that a strong correlation was found only for <60% of the patients. But these questions could be advanced with future studies. Aside from this concern, this is high quality and interesting paper and I would recommend publication.

Round 2

Reviewer 1 Report

All issues have been adressed sufficiently. I have no more remarks and I want to congratulate the authors on their hard work and their results.